# Evolutionary Patterns of Sex-Biased Genes in Three Species of Haplodiploid Insects

**DOI:** 10.3390/insects11060326

**Published:** 2020-05-26

**Authors:** Yu-Jun Wang, Hua-Ling Wang, Xiao-Wei Wang, Shu-Sheng Liu

**Affiliations:** Ministry of Agriculture Key Laboratory of Agricultural Entomology, Institute of Insect Sciences, Zhejiang University, Hangzhou 310058, China; wangyujun@zju.edu.cn (Y.-J.W.); wang_hual@126.com (H.-L.W.); xwwang@zju.edu.cn (X.-W.W.)

**Keywords:** gene expression, whitefly, *Bemisia tabaci*, RNA-seq, transcriptome

## Abstract

Females and males often differ obviously in morphology and behavior, and the differences between sexes are the result of natural selection and/or sexual selection. To a great extent, the differences between the two sexes are the result of differential gene expression. In haplodiploid insects, this phenomenon is obvious, since males develop from unfertilized zygotes and females develop from fertilized zygotes. Whiteflies of the *Bemisia tabaci* species complex are typical haplodiploid insects, and some species of this complex are important pests of many crops worldwide. Here, we report the transcriptome profiles of males and females in three species of this whitefly complex. Between-species comparisons revealed that non-sex-biased genes display higher variation than male-biased or female-biased genes. Sex-biased genes evolve at a slow rate in protein coding sequences and gene expression and have a pattern of evolution that differs from those of social haplodiploid insects and diploid animals. Genes with high evolutionary rates are more related to non-sex-biased traits—such as nutrition, immune system, and detoxification—than to sex-biased traits, indicating that the evolution of protein coding sequences and gene expression has been mainly driven by non-sex-biased traits.

## 1. Introduction

Females and males often differ obviously in morphology and behavior, and most of the differences between sexes are the result of natural selection and/or sexual selection [1]. While females and males exhibit different phenotypes, they are roughly the same at the genetic level, indicating that the differences between sexes are largely the result of differences in gene expression. These genes are subject to natural selection and sexual selection pressures from both sexes and are sometimes even subject to conflicting selection pressures [2]. Sex-biased genes contain genes that only express in one sex (sex-specific expression) or show higher expression in one sex than in the other (sex-enriched expression). Both the coding sequences and the expression levels of sex-biased genes show rapid evolution in diploids, such as *Drosophila*, mammals, and birds [1,3,4,5,6]. The evolution of the expression of sex-biased genes may display a positive correlation with that of protein sequences [7].

Approximately 15% of animals are haplodiploids; some of them are social insects, like ants and honey bees, and others are non-social solid haplodiploid insects [8,9]. Haplodiploid males develop from unfertilized zygotes and females from fertilized zygotes. Therefore, their chromosomes are the same, except that females have two sets while males have only one. This means that the differences between the two sexes are mainly the results of differential gene expression [1]. In contrast to diploids, haplodiploid males have no paternal origin, and all their genes are derived from the mothers. In addition, haplodiploid males pass their genes only to the females of next generation. For a given gene, two-thirds of the copies are in the female, and one-third in the male. In haplodiploids, when a gene is deleterious to the male, it will not be influenced by the allele in the male and thus is exposed directly to the pressure of natural selection. Therefore, both explicit and implicit deleterious mutations have lower frequencies in haplodiploids than in diploids [10]. Theoretical studies show that the region of stability polymorphism in haplodiploids is smaller than that in diploids when differences in fitness or selection occur in opposite directions between the two sexes, leading to a higher capacity for sex-biased genes to evolve fast in haplodiploids [9]. In social haplodiploid insects, sex-biased genes—e.g., genes that are differentially expressed between queens and males—also show rapid evolution in expression and protein coding sequences [11,12,13].

We focus on a group of non-social haplodiploid insects, i.e., whiteflies of the *Bemisia tabaci* cryptic species complex that contains > 35 morphologically indistinguishable but genetically divergent species [14,15,16,17,18]. This species complex exhibits a cosmopolitan distribution and contains some important pests of a wide range of crops. For example, two of the whitefly species, tentatively named as Middle East Asia Minor 1 (MEAM1) and Mediterranean (MED), are notorious invasive pests [16]. The first large-scale global invasion of MEAM1 occurred in the late 1980s, and the invasion has been expanding worldwide since then. Since the early 2000s, a large-scale invasion of MED has been observed in many regions of the world [16,19]. Both species have been displacing indigenous whiteflies in many regions of invasion. For example, in China, the indigenous species Asia II 3, which causes little damage to crops, has been displaced by MEAM1 and/or MED in many areas [16,19,20]. One major factor involved in the displacement of Asia II 3 by MEAM1 has been identified as sexual behavioral interactions between the two species [21]. Another factor that may be involved in the displacement is the viruses they vector; compared to indigenous whitefly species, the two invasive species are more likely to benefit from their interactions with the viruses [22,23,24]. MED has developed a high resistance to major classes of insecticides, and in habitats where insecticide application is frequent, MED can displace MEAM1 [20]. Previously, pairwise comparisons of MEAM1, MED, and Asia II 3 were analyzed at both sequence and gene expression levels, suggesting that the genes related to metabolism and detoxification are highly divergent [25,26,27]. However, whether these differences between whitefly species are influenced by sex-biased genes remains largely unknown.

In this study, to investigate whether sex-biased genes evolve rapidly in protein coding sequences and gene expression, firstly we analyzed female and male transcriptomes of three species of the *B. tabaci* whitefly complex, i.e., MEAM1, MED, and Asia II 3. Next, we analyzed the relative contribution of sex and species to the divergence of expression in orthologous genes. We then analyzed the functional enrichment of fast-evolving genes in sex-biased categories to examine which molecular functions drive the evolution of protein coding sequences and gene expression.

## 2. Materials and Methods

### 2.1. Whitefly Cultures and RNA Isolation

Two MEAM1 whitefly colonies (*mtCOI* GenBank accessions: GQ332577 and KM821540), one MED whitefly colony (*mtCOI* GenBank accession: GQ371165), one Asia II 3 whitefly colony (*mtCOI* GenBank accession: DQ309076), and one greenhouse whitefly (*Trialeurodes vaporariorum*) colony were maintained on cotton *Gossypium hirsutum* (Malvaceae) cv. Zhe-Mian 1793 in a climate chamber controlled at 27 ± 1 °C, a photoperiod of 14 h light: 10 h darkness, and a 70% ± 10% relative humidity. The purity of each whitefly colony was monitored by the gene sequencing of *mtCOI*. We took female and male samples from whitefly colonies with two biological replicates for each sample. In total, 20 samples of whiteflies were tested (five colonies × two sexes × two replications). To determine their sex, adult whiteflies (1–5 days post-emergence) were placed into 5 × 0.5 cm glass tubes, one adult per tube, and then observed under a stereo microscope. Approximately 150 female adults or 200 male adults were collected for each sample. The total RNA of each sample was isolated using an SV total RNA isolation system (Promega) according to the manufacturer’s protocol. Each of the RNA samples of either female or male was further divided into two sub-samples. One of the two sub-samples of a given sex was mixed with a sub-sample of the opposite sex to obtain a sample of both sexes, which was then used to conduct Illumina sequencing to get a high-quality reference transcriptome sequences database to perform a de novo assembly and gene annotation. The other RNA sub-sample, either female or male, was used separately to conduct Illumina sequencing for obtaining the female and male RNA sequences respectively.

### 2.2. RNA-Sequencing, de novo Assembly for Reference Transcriptome Sequences, and Mapping for Samples

To attain reference sequences of high quality for the three species of *Bemisia* whiteflies, sequencing libraries were generated with mixed female and male RNA samples of each species using the NEBNext Ultra RNA Library Prep Kit for Illumina (NEB, Ipswich, MA, USA). The libraries were sequenced on an Illumina Hiseq 2000/2500 platform in Novogene Bioinformatics Technology Co., Ltd (Beijing, China). After removing reads containing adapter, ploy-N, or low-quality reads, a de novo assembly was carried out using Trinity software [28] with the default parameters for each species. The assembled sequences are available at the NCBI (National Center for Biotechnology Information) Transcriptome Shotgun Assembly (TSA) with the accession numbers GIBX00000000, GARQ00000000, and GICC00000000. The assembled sequences were used for annotation by Blastx against the NR (Non-redundant Protein Sequence Database), Swiss-Prot, and KEGG (Kyoto Encyclopedia of Genes and Genomes) databases with an E-value cut-off of E-5. The functional annotation of Gene Ontology (GO) terms was analyzed by the Blast2GO software [29]. The coding sequences of unigenes were predicted according to NR annotations.

Furthermore, to obtain the female and male RNA sequences for each of the species, sequencing libraries of 20 RNA samples total were generated using the NEBNext Ultra RNA Library Prep Kit for Illumina (NEB, USA). Each library was sequenced on an Illumina Hiseq 2000/2500 platform in Novogene Bioinformatics Technology Co. Ltd. After removing low-quality reads, clean data of all the samples were submitted to the NCBI SRA (Sequence Read Archive) database in BioProject PRJNA545218. RSEM (RNA-Seq by Expectation Maximization) [30] was used to map the reads to the corresponding transcriptome sequences of each species and estimate the transcript abundance of each gene (fragments per kilobase of exon per million fragments, FPKM). A differential expression analysis within the same species was conducted using edgeR [31]. Two-fold differences in expression and an FDR (False Discovery Rate) adjusted of *p* < 0.05 were used as the threshold of sex-biased genes.

### 2.3. Identification of Orthologs and Analysis of Sequence Evolution

Firstly, the predicted protein sequences of the whiteflies were combined and an all-vs-all blast (*p* < E-5) was performed using blastp. Next, the orthologous gene groups were identified using the OrthoMCL software with default parameters [32]. Only groups containing one gene from each of the species were reserved as 1:1 ortholog groups. A custom Perl script was used to handle multiple comparisons of the orthologous genes in ClustalW [33] and to extract the corresponding transcriptome fragments. Short fragments of ortholog groups (< 150 bp) were excluded from further analysis. The pairwise rate ratio of synonymous to non-synonymous substitutions (Ka/Ks) was calculated using the M0 model incorporated into PAML 4 (Phylogenetic Analysis by Maximum Likelihood) [34].

### 2.4. Normalization of Gene Expression Across Species

To compare the expression across different species and sexes, we further normalized the gene expression levels (FPKM) using a linear regression model:*y_i_* = *βx_i_* + *ε*,
where *x_i_* is the average gene expression (FPKM) across samples of one species in gene *i*, and *y_i_* is the average gene expression (FPKM) across samples of another species in the corresponding gene *i*. The standardized FPKM (sFPKM) was produced using the parameter *β* as a scaling factor for normalizing the gene expression across each species by adding a pseudocount of 0.01. Then, the orthologous genes with an average sFPKM < 2 were further filtered.

### 2.5. Analysis of the Global Pattern of Gene Expression Levels

A principal component analysis of sFPKM was conducted using the “prcomp” function in the “stats” package in R (http://www.R-project.org/). Gene expression phylogenies were constructed based on pairwise distance matrices (1-ρ, ρ was Spearman’s correlation coefficient) of sFPKM [35]. We tried the NJ (neighbour-joining), BIONJ and FastME approaches provided by the “ape” package [36] in R project, and found that the tree constructed using the balanced FastME approach [37] was more symmetrical within the tree branch between female and male than those constructed using the other two approaches.

### 2.6. Analysis of Gene Expression among Species and Sexes

An analysis of variance (ANOVA) was used to decompose the source of variation. For each gene, an ANOVA model was built in which the expression was the sum of sex, species, the interaction of sex and species, and the residual term:*y_ij_* = *μ* + *sex_i_* + *species_j_* + *sex_i_* × *species_j_* + *ε_ij_*,
where *y_ij_* is the expression of one gene in *sex**_i_* and *species**_j_*, μ is the basal expression levels of the gene, *sex**_i_* is the coefficient for *sex**_i_*, *species**_j_* is the coefficient for *species**_j_*, and *ε**_ij_* is the residual term. The ANOVA model was implemented using the function lm from the “stats” package in R project, followed by a Tukey’s HSD (Honest Significant Difference) test using the function “TukeyHSD” from the “stats” package in R project. A custom script was modified and used for this ANOVA approach [38].

The expression variation in each orthologous gene across sexes and species was calculated using the method of Ometto et al. [17] for comparisons between fire ants:$$\mathrm{Vx}=\left(\log_{2}\left|\frac{X_{\mathrm{pop}\;1\;}^{\mathrm{female}}}{X_{\mathrm{pop}\;2}^{\mathrm{female}}}\right|+\log_{2}\left|\frac{X_{\mathrm{pop}\;1\;}^{\mathrm{male}}}{X_{\mathrm{pop}\;2}^{\mathrm{female}}}\right|\right)/2\;\mathrm{Dx}=\left(\log_{2}\left|\frac{X_{\mathrm{MEAM}\;1\;}^{\mathrm{female}}}{X_{\mathrm{MED}}^{\mathrm{female}}}\right|+\log_{2}\left|\frac{X_{\mathrm{MEAM}\;1\;}^{\mathrm{female}}}{X_{\mathrm{Asia}\;\mathrm{II}\;3}^{\mathrm{female}}}\right|+\log_{2}\left|\frac{X_{\mathrm{Asia}\;\mathrm{II}\;3\;}^{\mathrm{female}}}{X_{\mathrm{MED}}^{\mathrm{female}}}\right|+\log_{2}\left|\frac{X_{\mathrm{MEAM}\;1\;}^{\mathrm{male}}}{X_{\mathrm{MED}}^{\mathrm{male}}}\right|+\log_{2}\left|\frac{X_{\mathrm{MEAM}\;1\;}^{\mathrm{male}}}{X_{\mathrm{Asia}\;\mathrm{II}\;3}^{\mathrm{male}}}\right|+\log_{2}\left|\frac{X_{\mathrm{Asia}\;\mathrm{II}\;3\;}^{\mathrm{male}}}{X_{\mathrm{MED}}^{\mathrm{male}}}\right|\right)/6\;\mathrm{Sx}=\left(\log_{2}\left|\frac{X_{\mathrm{MEAM}\;1\;}^{\mathrm{male}}}{X_{\mathrm{MEAM}\;1}^{\mathrm{female}}}\right|+\log_{2}\left|\frac{X_{\mathrm{MED}\;}^{\mathrm{male}}}{X_{\mathrm{MED}}^{\mathrm{female}}}\right|+\log_{2}\left|\frac{X_{\mathrm{Asia}\;\mathrm{II}\;3\;}^{\mathrm{male}}}{X_{\mathrm{Asia}\;\mathrm{II}\;3}^{\mathrm{male}}}\right|\right)/3$$
Rx = Dx/Vx,
where pop1 and pop2 represent the two populations of MEAM1. X is an average of sFPKM within a biological replicate across each gene.

## 3. Results

### 3.1. Summary of RNA Sequencing and Mapping

To attain reference sequences of high quality for downstream analysis, three novel versions of the MEAM1, MED, and Asia II 3 transcriptome sequences were generated with mixed female and male RNA samples of each species using the Illumina platform. After filtering out low quality and adapter-related sequences, about 181 million, 102 million, and 189 million 100 bp pair-end clean reads of the three species were obtained (Appendix A). These reads were assembled using Trinity software [28], resulting in 63,020, 46,805, and 59,120 unigenes with N50 (the minimum unigene length need to cover 50% of the transcriptome) lengths of 1719, 2319, and 1823 bp, respectively (Appendix A). For functional annotations, all of the unique sequences were blasted against NR using Blastx, and 12,794, 12,795, and 14,408 unigenes of the three species were annotated (Appendix A). In addition, 8193, 15,223, and 8194 unigenes were used for a GO annotation for MEAM1, MED, and Asia II 3, respectively.

In each of the 20 samples across different species and sexes (female and male samples from two colonies of MEAM1, one colony of MED, one colony of Asia II 3, and one colony of greenhouse whitefly, with two biological replications; 5 colonies × 2 sexes × 2 replications), 9.7–14.4 million RNA-seq single-end reads were generated. These reads were then mapped to the de novo transcriptome sequences of each of the four species, respectively. The mapping rate of the three *Bemisia* species ranges from 87.5% to 89.9%, while that of the greenhouse whitefly ranges from 67.6% to 70.9% (Appendix A). Biological replicates showed high reproducibilities (Pearson correction of > 0.85 except 1 pair; Appendix A), indicating the reliability of the data.

### 3.2. Analysis of Sex-Biased Genes

Sex-biased genes in the three *Bemisia* species were analyzed with the threshold of a two-fold difference in expression between females and males with an FDR-adjusted *p*-value of 0.05. In all transcriptome sequences of the three species, 14–17% of the genes were female-biased and 11–16% were male-biased (Appendix A). In the genes with NR annotations, compared with all genes, the numbers of female-biased genes are over-represented (all hyper-test *p* < 0.05), while the numbers of non-sex-biased are under-represented (all hyper-test *p* < 0.05) (Appendix A). To examine the biological functions associated with the more active expression of sex-biased genes, we performed GO and KEGG enrichment analyses. In the GO analyses, the female highly expressed genes were enriched in nucleic acid binding, translation, and RNA binding; in the KEGG pathways, the genes associated with RNA transport and spliceosome, nucleotide excision repair, and cell cycle were more actively expressed. Furthermore, female-biased genes with a > 8-fold expression are associated with oocyte maturation, cell cycle, and the meiosis pathway, indicating that these genes are related to egg development. Male highly expressed genes were enriched in a structural constituent of the cuticle, calcium signaling pathway, G-protein coupled with receptor activity, and neuroactive ligand-receptor interaction, indicating that male-biased genes are more closely associated with nerves and senses (Appendix A)

### 3.3. Evolution of Coding Sequences across Species

Generally, the ratio of non-synonymous (Ka) to synonymous (Ks) substitutions in closely related species can be used to illustrate the evolution of protein coding sequences. In MEAM1, MED, and Asia II 3, 5490 1:1 orthologous genes were obtained using orthoMCL [30]. In these 1:1 ortholog groups, the Ka/Ks ratio of 4946 groups could be calculated for pairwise sequence comparisons. In the three pairs of MEAM1-MED, MEAM1-Asia II 3, and MED-Asia II 3, 52% of the pairwise groups show a Ka/Ks < 0.1, whereas 98, 65, and 63 groups show a Ka/Ks > 1 (Appendix A). Different ranges of Ka/Ks were used in the GO and KEGG enrichment analyses to examine whether fast evolving genes are enriched in some gene functions. In the low Ka/Ks category (average Ka/Ks < 0.1), genes related to ribosome, phagosome, and some signaling pathways are enriched (Appendix A). Meanwhile, in the high Ka/Ks category (average Ka/Ks > upper quartile, 0.227), genes related to lysosome, nutrient digestion, and some metabolism pathways (oxidative phosphorylation, glycerophospholipid metabolism, aminoacyl-tRNA biosynthesis, vitamin digestion, and absorption) are enriched. These genes mostly play essential roles in digestion and may be associated with the capacity in adapting to host plants (Appendix A).

### 3.4. Global Pattern of Gene Expression across Sex and Species

To obtain the overall pattern of gene expression, a phylogeny analysis and principal component of expression were conducted for all the orthologous genes. The expression of 1:1 orthologs from the three *Bemisia* species and the outgroup greenhouse whitefly were constructed to a rooted tree based on pairwise distance matrices (1-ρ, Spearman’s correlation coefficient) (Figure 1a and Appendix A). The outgroup departs first and then the females and males move apart into two groups. The variation in gene expression between the three *Bemisia* species and the greenhouse whitefly is larger than the variation between sexes, indicating that the expression pattern of the greenhouse whitefly differs significantly from that of the three *Bemisia* species. This seems reasonable, as the *Bemisia* species and the greenhouse whitefly belong to different genera. The expressions of 1:1 orthologs from the three *Bemisia* species were also constructed as an unrooted tree, which appears similar to that of the rooted tree (Figure 1b and Appendix A). On the whole, males have longer branches than females, indicating that the difference in gene expression between males is larger than that between females. These patterns of gene expression are also shown by the results of the principal component analysis (Figure 1c,d). The variation in gene expression between the three *Bemisia* species was clustered preferentially by species, indicating that these species are closely related and share similar gene expression patterns in the same sex, while the females and males in each species differ in expression patterns although the two sexes share the same genome.

### 3.5. Evolution of Gene Expression across Species

Similar to the evolution of protein coding sequences, the evolution of gene expression levels can also be measured by the ratio of interspecific to intraspecific expression variation. Relatively low variations in both interspecific and intraspecific expression may indicate a purifying selection, which may be a result of high functional constraints; a relatively high interspecific variation and low intraspecific variation may indicate a positive selection, a result of the diversity of different species. A relatively high variation in both interspecific and intraspecific may indicate a relaxed selection, an outcome of the weak constraint of gene expression [1,13]. The interspecific variation across the three *Bemisia* species (Dx); the intraspecific variation in gene expression across sexes in the two populations of MEAM1 (Vx); and their ratio, Rx, were calculated for each orthologous gene. In order to test whether genes with different possible selections are enriched in some functions, the enrichment of GO terms and KEGG pathways was analyzed. Genes with relatively high interspecific variation and low intraspecific variation (the upper 50% percentile for Dx and lower 50% percentile for Vx) are enriched in some immune pathways (peroxisome, apoptosis); some metabolism pathways (drug, retinol, etc.); the MAPK (Mitogen-Activated Protein Kinase) signaling pathway; and in male courtship behavior, brain development, and cell cycle (Appendix A). These genes may be under positive selection and tend to evolve fast in gene expression. Genes with a relatively low expression variability (in the lower 50% percentile for both Dx and Vx) are enriched in proteasome and phagosome, indicating that these genes may be under purifying selection (Appendix A). Compared to the evolution of protein coding sequences, fewer GO terms or KEGG pathways are enriched in both a high Ka/Ks ratio and high Rx ratio (antigen processing and presentation, lysosome, glycerophospholipid metabolism, and cysteine-type peptidase activity), indicating that some molecular processes are influenced by the evolution of protein coding sequences and gene expression.

### 3.6. Sex-Biased Genes in Orthologs Groups

We analyzed the distribution of sex-biased genes in orthologs. Of the 5490 orthologs, 2410, 2405, and 2321 are female-biased genes in MEAM1, MED, and Asia II 3, while 808, 742, and 650 are male-biased genes in the three species, respectively (Figure 2a). Notably, the proportion of male-biased genes is significantly decreased in MEAM1 and MED (genes contained the NR annotation as background, hyper-test *p* = 0.04 and *p* < E-5, respectively). The Spearman corrections of overall expression in 5490 orthologs are 0.863, 0.81, and 0.798 in the three species pairs of MEAM1-MED, MEAM1-Asia II 3, and MED-Asia II 3, respectively. Only 27 genes (0.49%) have switched to opposite sex-biased in one or two species, indicating that sex-biased genes are quite consistent and stable among orthologs. In total, 22.3% of the genes (1227) are altered between female-biased and non-sex-biased in one or two species, and 8.4% between male-biased and non-sex-biased genes (459), showing that numerous genes are divergent in the sex-biased categories across different species.

### 3.7. Evolution of Coding Sequences of Sex-Biased Genes

There is a hypothesis that sex-biased genes evolve fast in protein coding sequences, since these genes are more likely subject to different levels of natural selection between sexes [1]. In order to test whether sex-biased genes evolve fast in protein coding sequences in the three *Bemisia* species, we analyzed the pattern of the Ka/Ks ratio of each orthologous gene across different sex-biased categories. The Ka/Ks of orthologous genes suggested that male-biased genes evolve slower than female-biased or non-sex-biased genes in all the three *Bemisia* species, while female-biased and non-sex-biased genes show no significant difference in evolutionary rates between MEAM1 and MED, and non-sex-biased genes evolve faster than female-biased in Asia II 3 (Figure 2b and Appendix A). In addition, in the category of genes with an average Ka/Ks of > 1, the number of non-sex-biased genes is the highest (23, 20, 25 in MEAM1, MED, and Asia II 3, respectively). Furthermore, in the genes with a high average Ka/Ks ratio, non-sex-biased genes are over-represented (higher than the upper quartile of the average Ka/Ks ratio; hyper-test *p* < 0.001), while female-biased and male-biased genes are under-represented (hyper-test *p* = 0.02 and *p* < 0.01, respectively). In consideration of the variation between species, non-sex-biased genes contain more species-biased genes, with a higher Ka/Ks ratio than female-biased or male-biased genes (Figure 3a). Thus, the hypothesis that sex-biased genes are fast evolving in protein coding sequences may not be supported in whiteflies.

### 3.8. Expression Evolution of Sex-Biased Genes

To test whether sex-biased genes evolve fast in gene expression levels in the three *Bemisia* species, the pattern of interspecific variation (Dx), intraspecific variation (Vx), and their ratio (Rx = Dx/Vx) mentioned above were analyzed. The Dx of both male-biased and female-biased genes is lower than that of non-sex-biased genes (Mann–Whitney U test, both *p* < 0.01) (Figure 4a and Appendix A). The Vx of both male-biased and female-biased genes is higher than that of non-sex-biased genes (Mann–Whitney U test, both *p* < 0.001) (Figure 4b and Appendix A). Therefore, non-sex-biased genes have a higher Rx than female-biased or male-biased genes (Mann–Whitney U test, both *p* < 0.001) (Figure 4c and Appendix A). Furthermore, both high Dx genes (higher than the upper quartile of Dx) and high Rx genes (higher than the upper quartile of Rx) are over-represented in non-sex-biased genes (hyper-test all *p* < 0.0001) and under-represented in female-biased and male-biased genes (hyper-test all *p* < 0.0001). In consideration of the variation between species, non-sex-biased genes also contain more species-biased genes, with a higher Rx ratio than female-biased and male-biased genes (Figure 3b). These patterns indicate that non-sex-biased genes may be more frequently under positive selection than female-biased or male-biased genes.

### 3.9. Differential Expression across Sexes and Species

In orthologous genes, the variations in gene expression may be the result of different sexes or different species or both. A two-way ANOVA with interactions was used to identify the source of variations in gene expression for each orthologous gene. The total gene expression variance for one orthologous gene across samples can be decomposed into four parts: sexes, species, both sexes and species, and residual variance. The ANOVA analysis was used to classify all the genes into five categories: sex-variable genes, species-variable genes, sex- and species-variable genes, interaction-variable genes (for these genes, the effect of species depends on the effect of sex or vice versa), and non-variable genes. Most genes are sex- and species-variable genes (1566, 32.2%), followed by sex-variable (1519, 31.3%), interaction-variable (1156, 23.8%), species-variable (441, 9.1%), and non-variable genes (178, 3.7%) (Figure 5a). In the sex-variable genes, 87.4% (1327) are sex-biased at least in one species, while 74.5% of sex- and species-variable genes (1167) are sex-biased at least in one species.

Sex-variable genes were further decomposed into female overexpressed or male overexpressed, and species-variable genes were further decomposed for pairwise comparisons (Appendix A). The results show that 16.4% of orthologs (796) are differentially expressed between MEAM1 and MED, 21.1% of orthologs (1025) are differentially expressed between MEAM1 and Asia II 3, and 21.7% of orthologs (1056) are differentially expressed between MED and Asia II 3. Sex-variable genes are enriched in gene transcription, translation, ribosomal activity, and proteasome activity (Appendix A). Species-variable genes are enriched in digestive gland secretion, nerve and immune aspects, while sex- and species-variable genes are enriched in signal transduction and digestive gland secretion (Appendix A). The evolution of protein coding sequences and gene expression levels was calculated for each category in order to determine the association between evolution and the source of variations. Sex-variable genes show a lower Ka/Ks, Dx, and Rx ratio than the median (all the Mann–Whitney U tests, *p* < 0.05) and a significantly higher Vx than the median (Mann–Whitney U test, *p* < 0.05) (Figure 5b–d and Appendix A). Species-variable genes show a higher Ka/Ks, Dx and Rx ratio than the median (all the Mann–Whitney U tests, *p* < 0.001), and their Vx is similar to the median (Mann–Whitney U test *p* > 0.05) (Figure 5b–d and Appendix A). Sex- and species-variable genes are under opposite effects from sex and species, with a median level of Ka/Ks and Rx (Figure 5b–d and Appendix A). This pattern suggests that the effect of sex might decrease the evolutionary rate in both the protein coding sequence and gene expression levels, while the effect of species may increase the evolutionary rate. This pattern appears consistent with the observation that sex-biased genes have a lower Ka/Ks and Rx, while non-sex-biased genes have a higher Ka/Ks and Rx, indicating that sex-biased genes evolve slower than non-sex-biased genes in both protein coding sequences and expression levels in the three *Bemisia* species.

### 3.10. Correlations of Sex-Biased Expression and Evolution of Sequences and Expression

There is a hypothesis that all measures of expression variation—including gene expression variation within species, among species, and between sexes—are associated with one another and the evolution of protein sequences and gene expression [39,40]. We tested this hypothesis in the three *Bemisia* species through analyzing whether the evolution indicators, Rx and Ka/Ks, and the gene expression variations between two sexes—(Sx), Vx, and Dx—are correlated with one another. Surprisingly, not all the measures are positively correlated, but the results could be divided into two clusters: one cluster includes Sx and Vx and the other includes Dx, Rx, and Ka/Ks. This kind of pattern indicates that Dx, Rx, and Ka/Ks may share a similar process of evolution, while Sx and Vx may be less constrained in the process of evolution, as mentioned above. Rx and Ka/Ks are positively correlated (ρ = 0.10, *p* < E-5), showing that the evolution of protein coding sequences and gene expression are correlated but the correlation is weak. The result is consistent with the outcome of the GO and KEGG enrichment of genes with a high Rx and high Ka/Ks ratio (Appendix A). Sx is positively correlated with Vx (ρ = 0.23, *p* < 0.001) and negatively correlated with Dx (ρ = −0.16, *p* < 0.0001) (Figure 6), suggesting that the gene expression variation between sexes may be subject to the same evolutionary processes as intraspecific variation, but may have different evolutionary processes from interspecific variation. Sx is negatively correlated with the evolution of sequences (ρ = −0.04, *p* < 0.0001) and expression (ρ = −0.44, *p* < 0.0001), a result consistent with that of the ANOVA analysis as well as the inference that non-sex-biased genes evolve fast in sequences and expression.

## 4. Discussion

### 4.1. Variation in Gene Expression Is More Influenced by Sex than by Species

The distance trees of gene expression across species and sexes and the principal component analysis show that variation in gene expression between the three *Bemisia* species is clustered preferentially by species but not by sex, suggesting that although females and males of whiteflies share the same genomes, they differ in expression patterns. Similar patterns were previously found in the comparisons between two fire ants, *Solenopsis invicta* and *S. richteri* [13]. The ANOVA analysis further separated the orthologous genes into different categories, showing that the number of sex-variable genes (63.5%) is higher than that of species-variable genes (41.3%) among the three *Bemisia* species, and that therefore the variation in gene expression is more influenced by sex than by species. The proportion of species-biased genes is lower than that observed between morphologically similar *D. melanogaster* and *D. simulans* (50%) [9], but higher than that (10%) observed between two closely related ants, *S. invicta* and *S. richteri* [13]. This suggests that the three whiteflies share a somewhat similar expression pattern and that a large number of genes that are differentially expressed among species may reflect some differential physiology and/or behavior traits.

### 4.2. Sex-Biased Genes Are Consistent among the Three Bemisia Species

We found that the number of male-biased and female-biased genes ranged from 11% to 17% in each of the three *Bemisia* species. Compared with a previous report of MEAM1 [36], the proportion of sex-biased genes is slightly higher. The enrichments of the GO and KEGG pathways are similar in these whiteflies, and the enriched categories of sex-biased genes are also similar; female-biased genes are enriched in DNA replication, RNA transport, and Ribosome activity and male-biased genes are enriched in the structural constituent of cuticles, a situation similar to that in a previous report on *Bemisia* whiteflies [41]. In the orthologous genes of the three *Bemisia* species, few genes (0.49%) are switched between female-biased and male-biased, while the others are consistent in the same sex-biased category or changed from sex-biased to non-sex-biased, a situation that was previously observed in a comparison of seven *Drosophila* species [6].

### 4.3. Sex-Biased Genes in Whiteflies Show a Pattern of Gene Sequences and Expressions Different from That in Social Haplodiploid and Diploid Organisms

A previous hypothesis states that sex-biased genes evolve fast in protein coding sequences and gene expression, since these genes are more likely to be subject to different levels of natural selection between sexes [1]. However, in the current study, we found that non-sex-biased genes predominantly have higher Ka/Ks ratios than male-biased genes as well as female-biased genes. These patterns differ from those in diploid animals with sex chromosomes and social haplodiploid insects. In diploid animals, a series of studies from *Drosophila*, mammals, and birds show that sex-biased genes tend to evolve fast [1,3,4,6,42,43,44]. Social haplodiploids, such as the fire ant *S. invicta* and the honeybee *Apis mellifera*, show a more complicated pattern because a large number of diploid females become sterile workers. For example, in protein coding sequences, the number of queen-biased genes in *S. invicta* pupal and adult stages is higher than that of unbiased genes, while genes exclusively expressed in workers seem to experience a relaxed selection [12].

In the three *Bemisia* species, non-sex-biased genes evolve fast not only in protein coding sequences but also in gene expression levels. They also show a higher interspecific expression variation and lower intraspecific expression variation, indicating that these genes have been under more positive selection and evolved relatively fast in gene expression in comparison with male-biased or female-biased genes. Furthermore, numerous non-sex-biased genes which show high interspecific variation and low intraspecific variation are species-biased. This pattern differs from that in diploids like *Drosophila* or social haplodiploid insects such as fire ants. In *Drosophila*, male-biased genes show a higher intraspecific expression variation and a higher ratio of interspecific to intraspecific variation than female-biased or non-sex-biased genes [42,45]. In fire ants, male-biased genes show a high interspecific and low intraspecific expression variation, indicating that these genes have been under stronger positive selection; worker-biased genes show high expression variations both within and between species, indicating that these genes have been under relaxed selection [13].

Another hypothesis states that all measures of expression variation—including gene expression variations within species, among species, and between sexes—are associated with one another and also with the evolution of protein sequences and gene expression in general, since these features may be subject to the same evolutionary processes [39,40]. In the orthologous genes of the three *Bemisia* species, the evolution of protein coding sequences is positively correlated with the evolution of gene expression, a situation consistent with that in the study of primates [46], *Drosophila* [6,7,47], and fire ants [40]. However, the correlation levels between the evolution of protein sequences and that of gene expression (ρ = 0.10) in these whiteflies were weaker than that of the species mentioned above. One possible reason for the relatively weak correlation may be that the gene expression levels in these whiteflies have adjusted in response to changes in both short-term stress and long-term adaptive evolution pressure in the environment; in this way, the transcriptional plasticity could help the whiteflies to adapt to changes in the environment, while the evolution of protein coding sequences reflects more on long-term adaptive evolution [48]. The variation between sexes is positively correlated with intraspecific variation but negatively correlated with interspecific variation, evolution of protein sequences, and evolution of gene expression in the three *Bemisia* species. In addition, using ANOVA to analyze each ortholog, we deposed total variance into two mean effects, species and sex, and found that the effect of sex may decrease the evolutionary rates of gene expression and protein coding sequences while the effect of species increases the evolutionary rates. This phenomenon differs from that in *Drosophila* [45] and fire ants [40], indicating that gene expression variation between sexes may exhibit a weak or even negative correlation with the evolution of protein coding sequences and gene expression.

### 4.4. Possible Causes for the Slow Evolution of Sex-Biased Genes

The phenotypic differences between females and males derive from the differences in gene expression between sexes, suggesting that these genes are influenced by two different directions between sex selection pressures or even by a conflict selection pressures between sexes [1]. Theoretical studies show that different fitnesses or even selections in opposite directions occur between the two sexes; the region of stability polymorphism in haplodiploids is smaller than that in diploids, and consequently, haplodiploids are likely to have more capacity to evolve fast in sex-biased genes [9]. However, our study shows a situation opposite to that derived from the theoretical studies. One possible reason is that some genetic, behavioral, and/or biological characters of the species have not been adequately considered in the theoretical model. Another possible reason is that, of the three *Bemisia* species analyzed in this study, two (MEAM1 and MED) are invasive and one (Asia II 3) is indigenous; genes related to non-sex-biased invasive characters like nutrition, immune system, and detoxification may evolve faster than the genes related to sex-biased factors, although sex-biased genes are influenced by conflict selection pressure between sexes, and are more likely to occur in adaptive evolution. This is consistent with the observation that the expressions of genes related to basic metabolism and detoxification are higher in the invasive species, MEAM1 and MED than in the indigenous species, Asia II 3 [25].

One existing hypothesis states that the high evolutionary rate of sex-biased genes may result from the expression of numerous genes in a limited number of tissues, such as ovary and testis [4,49], since tissue-specific expressed genes tend to evolve faster than broadly expressed genes [50,51]. In a comparison between human and chimpanzee as well as among four species of mice, testis-expressed genes showed a low intraspecific expression variation and a high interspecific expression variation [46,52]. In whiteflies, Ye et al. [53] found that 15 genes are under positive selection in the gut in the transcriptome comparison between MED and MEAM1. The lack of gene expression data from tissue-specific samples, especially from ovary and testis, hinders further understanding of the compounding factors that influence the evolutionary rate of sex-biased genes or expression breadth. Narrowly- expressed genes might serve as more important factors than sex-biased genes in driving the evolution of non-sex-biased genes.

In summary, this study shows that the patterns of expression variation are more influenced by sex than by species in the three species of haplodiploid whiteflies. The effect of sex may decrease the evolutionary rate, while the effect of species may increase it. Furthermore, in contrast to some hypotheses derived from theoretic reasoning, sex-biased genes of these haplodiploid insects do not evolve fast in either protein coding sequences or gene expression levels, a situation that differs from that in diploid animals and social haplodiploid insects. This difference may be attributed to some biological characteristics—such as nutrition, immune system, and detoxification—that is associated more closely with invasion than with sexual behavior in the two invasive whitefly species MEAM1 and MED.

## Figures and Tables

**Figure 1 insects-11-00326-f001:**
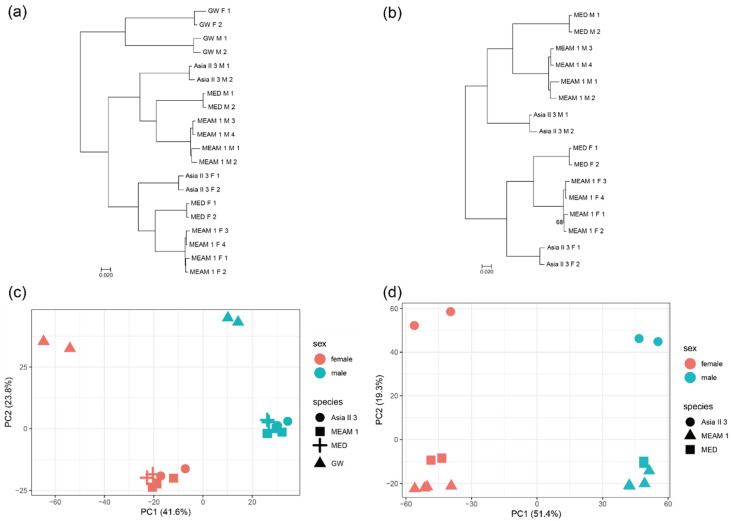
Global patterns of gene expression variation across species and sexes. Phylogeny trees of gene expression based on pairwise distance matrices (1-ρ, ρ is Spearman’s correlation coefficient) built by the balanced fastME method across the three *Bemisia* species and the outgroup greenhouse whitefly (GW) (**a**) and across the three *Bemisia* species (**b**) (F = female; M = male). In the nodes of phylogenetic trees, only the bootstrap values less than 90 were labeled. Phylogeny trees built by the neighbor-joining (NJ) method are presented in Appendix A. Results of the principal component analysis of gene expression across the three *Bemisia* species and greenhouse whitefly (**c**) and across the three *Bemisia* species (**d**).

**Figure 2 insects-11-00326-f002:**
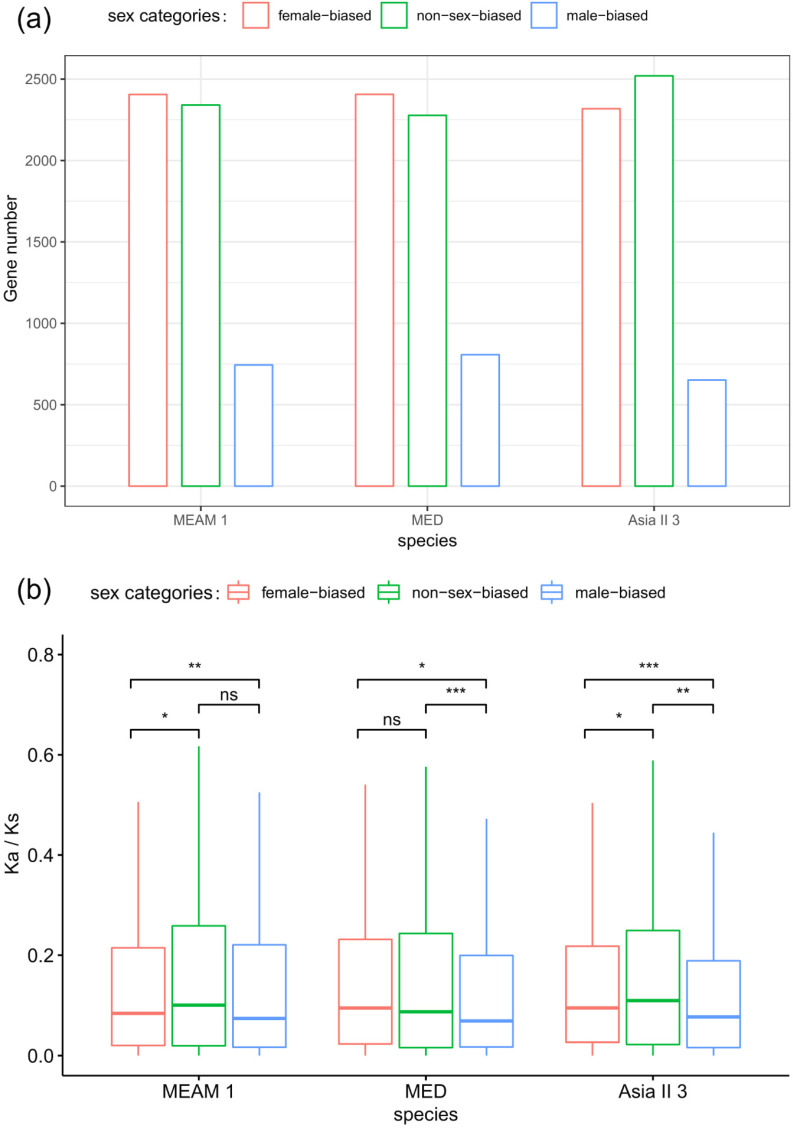
The number of orthologous genes that exhibit sex-biased or non-sex-biased expression and the evolution of their protein coding sequences. The number of orthologous genes (**a**) and average rate ratio of synonymous to non-synonymous substitutions (Ka/Ks) ratios for orthologous genes (**b**). Significant differences were determined by the Mann–Whitney U tests for pairwise comparisons: ns *p* > 0.05, * *p* < 0.05, ** *p* < 0.01, *** *p* < 0.001.

**Figure 3 insects-11-00326-f003:**
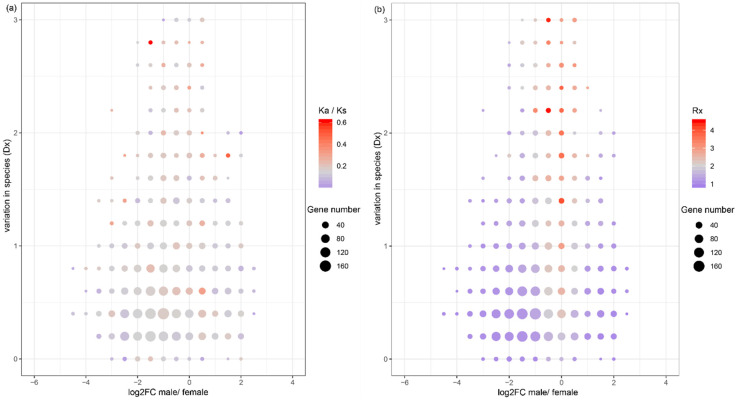
Evolution patterns in two-dimensional distribution of sex bias (log2FC male/female) and variation in species (Dx). The evolution of protein coding sequences is indicated by the Ka/Ks ratio (**a**) and the evolution of expression is indicated by the ratio of interspecific variation to intraspecific variation (Rx) (**b**); the color range represents the level of evolution indicators. The evolution indicators are grouped and the dot size indicates the number of genes in each group.

**Figure 4 insects-11-00326-f004:**
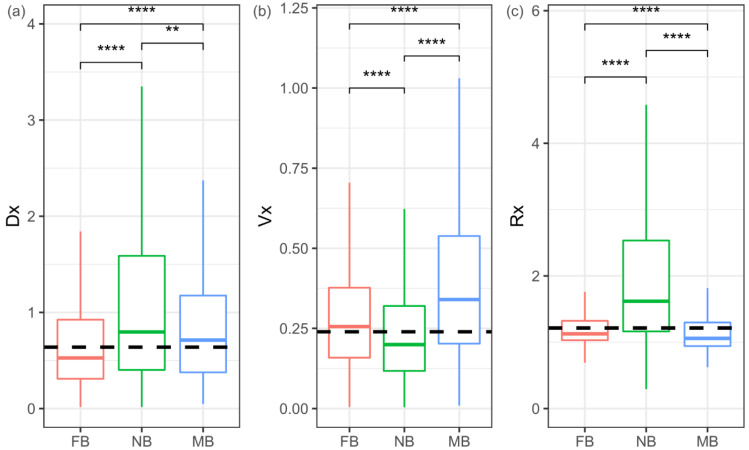
Evolution of gene expression in sex-biased and non-sex-biased genes. Gene expression variations: interspecific (Dx) (**a**), intraspecific (Vx) (**b**), and their ratio (Rx = Dx/Vx) (**c**) in each of the sex-biased categories. FB = female-biased; NB = non-sex-biased; MB = male-biased. Significant differences were determined using the Mann–Whitney U tests for pairwise comparisons: ** *p* < 0.01, **** *p* < 0.0001.

**Figure 5 insects-11-00326-f005:**
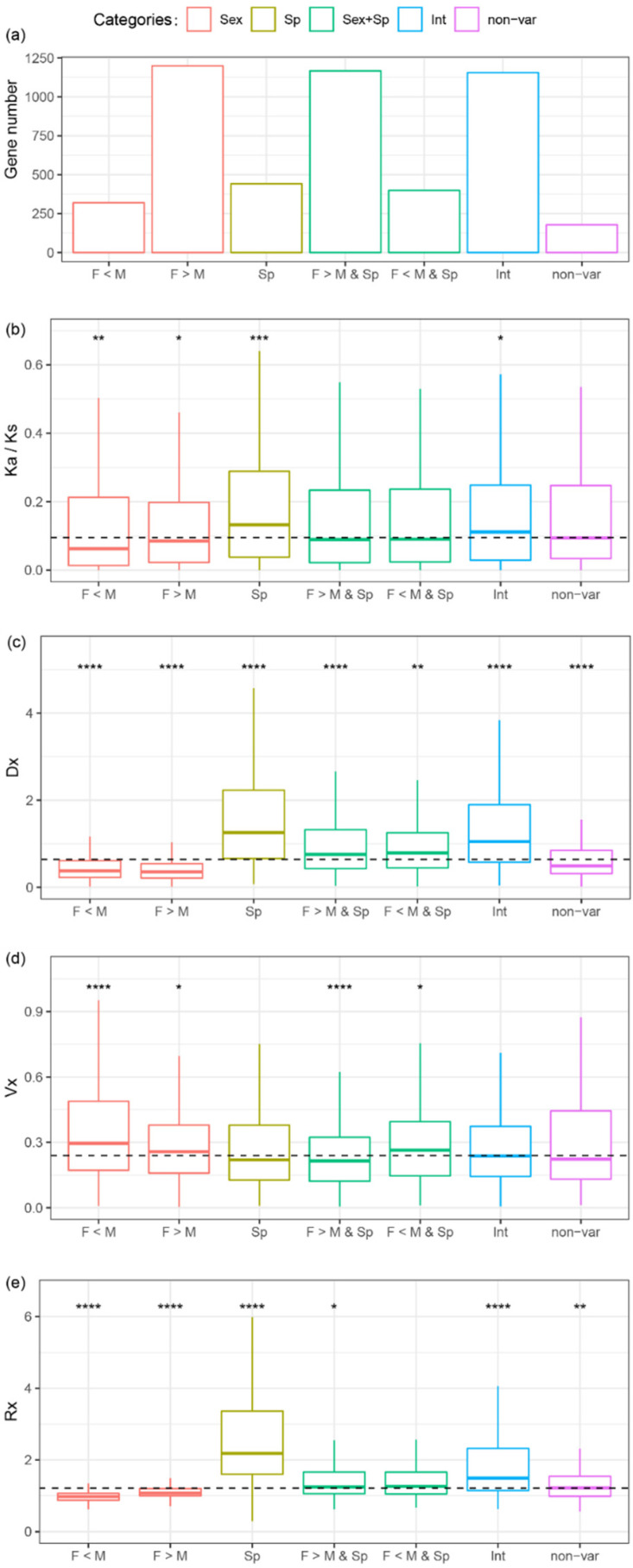
The evolution of protein coding sequences and gene expressions in sex-variable, species-variable, sex- and species-variable, interaction-variable, and non-variables genes. The number of genes (**a**), evolution of protein coding sequences (Ka/Ks ratio) (**b**), gene variations across species (Dx) (**c**), gene variations within Middle East Asia Minor 1 (MEAM1) (Vx) (**d**), and evolution of gene expression (Rx) (**e**) in each of the gene categories. Sex = sex-variable genes; Sp = species-variable genes; Sex + Sp = sex- and species-variable genes; Int = interaction-variable genes; non-var = non-variable genes; F = female-biased; M = male-biased. The dash line in the diagram (**b**), (**c**), (**d**), or (**e**) indicates the median of all values. Significant differences were determined using the Mann–Whitney U tests through the comparisons between each gene category and the median levels: * *p* < 0.05, ** *p* < 0.01, *** *p* < 0.001, **** *p* < 0.0001.

**Figure 6 insects-11-00326-f006:**
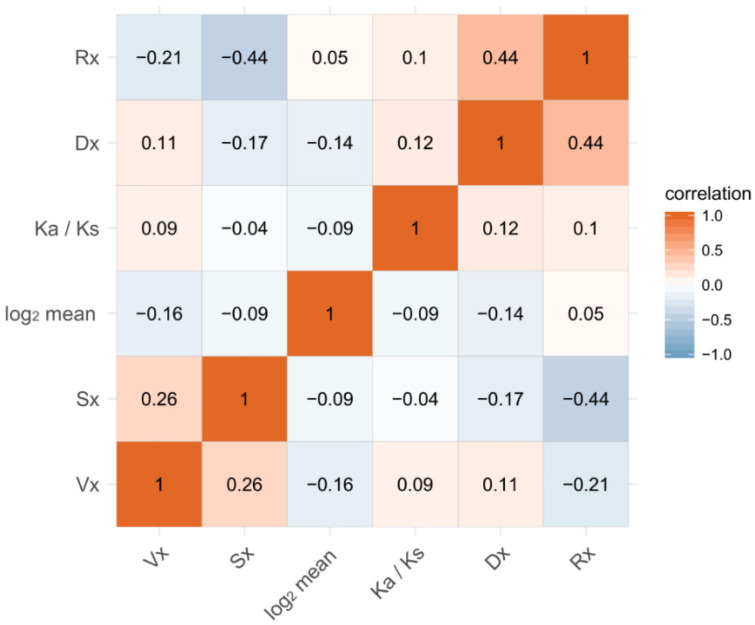
Correlations of gene expression measures and evolution of protein coding genes and gene expression. Spearman’s correlations for the pairwise measures were calculated, and all the correlations are significant (*p* < 0.05).

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
