# Peer review of "Evolutionary Patterns of Sex-Biased Genes in Three Species of Haplodiploid Insects"

_insects, 2020, doi:10.3390/insects11060326_

Round 1
Reviewer 1 Report
Wang et al., analyzed the evolutionary patterns of sex-biased genes in three species of haplodiploid insects. They report transcriptome profiles of males and females in three species of this whitefly complex finding that patterns of expression variation are more influenced by sex than by species in the three species of haplodiploid whiteflies. Before publication, I have some concerns shown below.
Major
1,
line 162, authors state that "transcriptome sequences were generated with MIXED female and male RNA samples of each species using Illumina platform."
While, line 179, authors state that "Sex-biased genes in the three Bemisia species were analyzed with the threshold of 2-fold difference in expression between females and males with an FDR adjusted p-value of 0.05."
So how did you know which RNA sequenced is from female or male by using the mixed female and male RNA samples? You must clearify this.
2,
Figs for phylogeny analysis, why don't you add bootstrap values in the trees in the manuscript?
Minor
In the methods, it may be better to combine 2.2 and 2.4, since both describe the methods of RNA sequencing.
Reviewer 2 Report
Authors studied the evolutionary patterns of sex-biased genes in three species of pest whiteflies, Bemesia. Whiteflies are haplodiploid insects, but do not live in (eu)sociality such as bees or ants. From social insects it is known that sex-biased genese evolve rather rapidly, such as in diploid insects or vertebrates.
The here presented results are of special interest, because authors found that in the Bemesia species, sex-biased genes of both sexes evolved much slower than non-sex-biased traits. Authors explain that with the unique life style of the (at least two) invasive pest species.
The study is carefully done and the manuscript is well written. My only major concern is that only two replications of each whitefly colony and sex were used for the experiments. At least three replications are usual.
Minors:
- Figure 2 and others: authors should explain only those numbers of asterisks, which are shown in the respective figure
- Headline 4.3: social haplodiploid and diploid organisms
- References: all species names have to be written in italics
Round 2
Reviewer 1 Report
Authors made an essential correction according to my questions. I am satisfied.
Author Response
Thanks for your comments.
Reviewer 2 Report
I still think that two repetitions per experiment are not enough.
Author Response
We explained on the issue of the number of replicates. We agree with the reviewer’s point of view and will take the reviewer’s advice in future experiments.